# CRMP2 as a Candidate Target to Interfere with Lung Cancer Cell Migration

**DOI:** 10.3390/biom11101533

**Published:** 2021-10-18

**Authors:** Xabier Morales, Rafael Peláez, Saray Garasa, Carlos Ortiz de Solórzano, Ana Rouzaut

**Affiliations:** 1Solid Tumors and Biomarkers Program, Center for Applied Medical Research, University of Navarra, Avda Pío XII, 55, 31008 Pamplona, Spain; xmorales@unav.es (X.M.); codesolorzano@unav.es (C.O.d.S.); 2Instituto de Investigación Sanitaria de Navarra (IdISNA), 31008 Pamplona, Spain; sgarasa@unav.es; 3Center for Biomedical Research of La Rioja (CIBIR), Neurodegeneration Area, Biomarkers and Molecular Signaling Group, Piqueras 98, 26006 Logroño, Spain; rpelaez@riojasalud.es; 4Division of Immunology and Immunotherapy, Center for Applied Medical Research (CIMA), University of Navarra, 31008 Pamplona, Spain; 5Department Biochemistry and Genetics, University of Navarra, Irunlarrea 1, 31080 Pamplona, Spain

**Keywords:** CRMP2, lung cancer, migration, cytoskeleton

## Abstract

Collapsin response mediator protein 2 (CRMP2) is an adaptor protein that adds tubulin dimers to the growing tip of a microtubule. First described in neurons, it is now considered a ubiquitous protein that intervenes in processes such as cytoskeletal remodeling, synaptic connection and trafficking of voltage channels. Mounting evidence supports that CRMP2 plays an essential role in neuropathology and, more recently, in cancer. We have previously described a positive correlation between nuclear phosphorylation of CRMP2 and poor prognosis in lung adenocarcinoma patients. In this work, we studied whether this cytoskeleton molding protein is involved in cancer cell migration. To this aim, we evaluated CRMP2 phosphorylation and localization in the extending lamella of lung adenocarcinoma migrating cells using in vitro assays and in vivo confocal microscopy. We demonstrated that constitutive phosphorylation of CRMP2 impaired lamella formation, cell adhesion and oriented migration. In search of a mechanistic explanation of this phenomenon, we discovered that CRMP2 Ser522 phospho-mimetic mutants display unstable tubulin polymers, unable to bind EB1 plus-Tip protein and the cortical actin adaptor IQGAP1. In addition, integrin recycling is defective and invasive structures are less evident in these mutants. Significantly, mouse xenograft tumors of NSCLC expressing CRMP2 phosphorylation mimetic mutants grew significantly less than wild-type tumors. Given the recent development of small molecule inhibitors of CRMP2 phosphorylation to treat neurodegenerative diseases, our results open the door for their use in cancer treatment.

## 1. Introduction

Lung cancer is the most common malignant tumor and the leading cause of cancer-related death worldwide; non-small cell lung cancer (NSCLC) is the most prevalent subtype. In 2020, there were 2.2 million tracheal, bronchial, and lung cancer cases globally, with 1.8 million deaths, accounting for approximately 17% of all cancer-related deaths [1]. This high mortality rate and low 5-year survival are mainly due to the early onset of metastases, which often happens even before the detection of the primary tumor [2].

For this reason, the study of the migratory and invasive capacity of NSCLC cancer cells and the mechanisms involved in this process, constitutes a matter of continuous research to provide new ways to improve patient prognosis.

NSCLC cells invade mostly through integrin-mediated mesenchymal migration. This process relies on the structuration of cellular protrusions, such as lamellipodia, filopodia, and invadopodia resulting from the coordinated activity of actin fibers with microtubules (MTs) and intermediate filaments. In particular, microtubules form highly dynamic scaffolds that give rise to and modify cellular protrusions [3].

MTs are hollow filaments composed of monomers of tubulin arranged in a polarized head-to-tail fashion. MTs constantly switch between phases of polymerization and depolymerization, in a process known as dynamic instability [4]. Abnormal regulation of these cytoskeleton structural proteins is frequent in cancer cells [5]. Therefore, drugs that depolymerize microtubules, such as Vinca alkaloids, or stabilize them as taxanes, have been proposed as potent anticancer agents [6]. These drugs can arrest aberrant cell division [7] and metastasis [8].

Several microtubule-associated proteins (MAPs) contribute to the elongation of the tubulin filaments [9]. Essential among them is the tubulin adaptor protein Collapsin Response Mediator Protein 2 (CRMP2), a ubiquitously expressed phosphoprotein best known for its affinity for tubulin heterodimers. CRMP2 regulates MT function by transporting tubulin dimers to the growing +TIP of the MT [10]. CRMP2 also interacts with proteins with varied functions, such as voltage-gated sodium and calcium channels, endocytic proteins, and several cytosolic protein kinases, such as ROCKII and RECK [11]. CRMP2 interaction with its partners is regulated by post-translational modifications. The most frequent one is phosphorylation in serine and threonine residues, but instances of other modifications such as SUMOylation, glycosylation or oxidation have also been described. In this sense, dysregulated post-translational modifications of CRMP2 contribute to diseases, such as cancer, neurodegenerative diseases, chronic pain, and bipolar disorder [12].

CRMP2 phosphorylation on Ser522 by the protein kinase Cdk5 and in the residues Thr509, Thr514 and Ser518 by GSK-3β diminish its affinity for tubulin [13]. Altered phosphorylation in these residues impedes correct tubulin polymerization and affects essential physiological functions such as cell division, vesicle transport, and cell migration. In this sense, cancer progression has been associated with alterations of CRMP2 phosphorylation in tumors of the lung, breast, prostate, and brain [14,15,16].

In this work, we specifically address the involvement of CRMP2 in the migration of lung carcinoma cells. Confocal microscopy-based analyses led us to find how CRMP2 localizes into the lamella of migrating cells in a phosphorylation-dependent manner. To this end, we have developed CRMP2 phosphorylation mutants in its residue Ser522, the target of Cdk5 kinase, and observed how these modifications affect CRMP2 localization during cell migration and tubulin stability. We also provide preliminary in vivo evidence on the importance of correct CRMP2 phosphorylation for tumor growth in a mouse model of lung adenocarcinoma. This topic is of particular interest as early targeting of cancer growth might prevent metastasis, the most ominous feature of tumors [1].

## 2. Materials and Methods

### 2.1. Cell Culture 

The human NSCLC cell lines A549 and H1299 were obtained from the American Type Culture Collection (ATCC, LGC-Promochem, Barcelona, Spain). Cells were grown at 37 °C, 5% CO_2_ in RPMI medium (Lonza, Barcelona, Spain) supplemented with 10% FetalClone III (Thermo Fisher Scientific, Waltham, MA, USA) and 100 units/mL penicillin-streptomycin. A549 and H1299 cell lines were authenticated by PCR amplification of genomic DNA, as previously described by Oliemuller et al. [17].

A549 phosphomimetic (Ser522 to Asp) and phosphodefective CRMP2 (Ser522 to Ala) mutants were obtained by mutagenesis, as previously described by Oliemuller et al. [17]. A549 endogenous CRMP2 expression was transiently silenced by transfecting 25 μM siRNA non-targeting control sequences (siNT-RNA) and CRMP2 siRNAs (siCRMP2- RNA) using ON-TARGETplus siRNA system offered by Horizon with the DharmaFect transfection system (Thermo Scientific, Waltham, MA, USA). This system uses a patented modification pattern for specificity that is combined with the SMARTselection algorithm for efficient target gene silencingTransfection with this siRNA was performed as previously described by Oliemuller et al. [17]. 

A549 cell line was stably transfected with the pLEGFP-CRMP2 (donated by Dr. Serrano, CNIO, Madrid, Spain), pLEGFP, plasmids using the X-tremeGENE 9 DNA transfection kit (Roche, Mannheim, Germany) according to the manufacturers’ protocol. Clones were selected by culturing cells in the presence of 1 mg/mL G418.

### 2.2. Antibodies and Reagents

Nocodazole and Geneticin (G418) were purchased from Sigma-Aldrich (Steinheim, Germany). Recombinant Human Exodus-2/CCL21 was purchased from PeproTech (London, UK). Clathrin-coated vesicle inhibitor chlorpromazine was purchased from Santa Cruz Biotechnology (Dallas, TX, USA). MMP inhibitor GM6001 was purchased from Millipore (Billerica, MA, USA). Recombinant Human EGF was obtained from R&D Systems (Minneapolis, MN, USA).

Antibodies: α-tubulin (clone DM1A), β1 integrin (clone BV7), pCRMP2 T514, GADPH and conjugated Alexa 594-conjugated transferrin were purchased from Abcam (Cambridge, UK). CRMP2 (C-terminal region) and pCRMP2 S522 were purchased from ECM Bioscience (Versailles, KY, USA). Detyrosinated tubulin was purchased from Millipore (Billerica, MA, USA). β1 integrin (clone 12G10), EB1 (clone KT51) and IQGAP1 (clone D-3) were purchased from Santa Cruz Biotechnology (Dallas, TX, USA). β-actin and γ-tubulin were purchased from Sigma-Aldrich (Steinheim, Germany). Immunofluorescence conjugated secondary antibodies Alexa Fluor 488, 594 and 647 donkey anti-mouse, donkey anti-rabbit or donkey anti-rat were purchased from Invitrogen (Barcelona, Spain). Western blot HRP-conjugated anti-mouse, anti-rabbit or anti-rat IgG secondary antibodies were from Santa Cruz (Dallas, TX, USA). Isotype controls mouse IgG1, rabbit IgG, and rat IgG2a were purchased from Biolegend (London, UK).

### 2.3. Cell Adhesion and Migration Assays

Adhesion of A549 cells was performed in 96-well flat-bottom cell culture plates (Sigma Aldrich, Steinheim, Germany). The wells were coated with collagen type I at 50 μg/mL, fibronectin at 10 μg/mL, collagen IV at 50 μg/mL, vitronectin at 1 μg/mL, and 3% BSA, used as negative adhesion control (Sigma-Aldrich, Steinheim, Germany) was performed as previously described [18].

Cell migration assays were performed as described previously [18]. Briefly, 5 × 10^4^ cells were placed in the upper compartment of an 8 μm pore-sized Boyden insert (Corning, NY, USA) covered with 50 μg/mL collagen type I. A549 and H1299 cell migration was stimulated by filling the lower compartment of the chamber with 20% serum-supplemented RPMI culture media. Three membranes of each cell type and condition were analyzed, in which four representative fields were acquired per Boyden membrane (number of images, *n* = 12). Data were plotted as the number of migrating cells normalized to the A549 control cell line. 

To evaluate cell migration, through 3D matrices we followed already published protocols [19]. In this case A549 and H1299 cell lines were embedded in Matrigel (2.5 mg/mL) at a concentration of 1000 cells/μL, and placed in Boyden inserts. Cell invasion was stimulated by filling the lower compartment of the chamber with 20% serum-supplemented RPMI culture media for 48 h. Cells the in the lower compartment were subsequently fixed, stained, and quantified as described above. Data were normalized relative to the A549 or H1299 non-transfected cell line.

### 2.4. Kinetic Parameters 

The kinetic parameters of the A549 cell line were analyzed based on the wound closure assay. Briefly, 2.5 × 10^5^ cells were plated on a 24 wells plate (Corning, New York, USA) and cultured to confluence with RPMI culture media supplemented with 10% serum at 37 °C. Cells were serum-starved for 16 h before scratching the cell monolayer using a pipette tip. Cell migration was stimulated by placing 200 ng/mL CCL21 (PeproTech, London UK). Subsequently, wound closure was recorded by confocal video-microscopy every 10 min for 12 h using a Cell Observer SD Spinning disk inverted confocal microscope (Zeiss, Jena, Germany) equipped with a 10X N-Achroplan objective (N.A. 0.25).

Individual cell trajectories were segmented using the Manual Tracking plugin developed for ImageJ. The velocity (μm/min) and directness (ratio of Euclidean to accumulated distance) plots were quantified from the tracking data obtained from ImageJ using the Chemotaxis and Migration Tool software (Ibidi, Martinsried, Germany). A random movement has a directness coefficient value of zero, while a fully oriented migration has the maximum directionality value, which is one. Three videos of each cell type and condition were analyzed, in which 20 individual cells were tracked. Data were normalized relative to the A549 control cell line.

### 2.5. Flow Cytometry

To evaluate the expression of β1 integrin in the membrane of A549 cells, 2 × 10^5^ cells were incubated for 20 min on ice with one μg/mL of anti-β1 integrin antibody (12G10, Santa Cruz, Dallas, TX, USA) or a non-specific mouse IgG1 isotype control (Biolegend, London, UK). After primary antibody incubation, cells were washed with cold flow cytometry buffer (1% BSA, EDTA 2 mM, 0.1% sodium azide in PBS) and incubated for 10 min on ice with the secondary Alexa Fluor 488 donkey-anti-mouse antibody (Invitrogen, Barcelona, Spain). Subsequently, excess secondary antibody was washed off, and cells were resuspended in flow cytometry buffer. Fluorescence measurements and data analysis were performed using BD FACSCalibur (BD Bioscience, San Jose, CA, USA). Three independent replicas were performed for each cell type and condition. At least 5 × 10^4^ cells were analyzed per experiment.

For β1 integrin uptake cells were serum-starved for 1 h and then incubated for 20 min on ice with anti-β1 integrin antibody (12G10, Santa Cruz, Dallas, TX, USA). The excess antibody was washed three times in a cooled flow cytometry buffer. Subsequently, cells were incubated for 10 min on ice with the secondary Alexa Fluor 488 donkey-anti-mouse antibody. Internalization of β1 integrin was stimulated by incubating cells in pre-warmed serum-free media for 5-, 15-, and 30 min. The non-internalized antibody was removed by washing with an acidic buffer (0.5 M NaCl, 0.2 M acetic acid, pH 2.5). When needed, clathrin-coated vesicle uptake was inhibited with 50 μM chlorpromazine for 30 min at 4 °C. β1 integrin internalization was analyzed by flow cytometry as described above. Data were plotted as the percentage of internalized β1 integrin relative to its expression in the cell membrane at 4 °C. 

For β1 integrin recycling analysis, integrin uptake was performed for 30 min in serum-free media as described above. Once β1 integrin was internalized, cells were chilled to block vesicle trafficking, and the non-internalized antibody was removed by washing with an acidic buffer. β1 integrin recycling was stimulated for 30 min with pre-warmed media supplemented with 10% serum. At this point, the externalized integrin was labeled for 10 min at 4 °C with the secondary Alexa Fluor 488 donkey-anti-mouse antibody. The externalized β1 integrin was analyzed on a FACScan flow cytometer using Cellquest software (BD Bioscience, San José, CA, USA). Data were plotted as the percentage of recycled β1 integrin relative to the integrin uptake after 30 min at 37 °C.

### 2.6. Lamellipodia Protein Purification

Protein purification from lamellipodia was performed as described by Kemble et al. [20]. Briefly, 1 × 10^5^ cells were seeded in the upper compartment of a 3 μm pore-sized Boyden chamber insert (BD Biosciences, San Jose, CA, USA). Cell migration was stimulated by filling the lower compartment of the chamber with a 20% serum-supplemented RPMI culture media. After 20-, 40-, 60-, and 80 min cells were fixed in pre-cooled methanol, and the upper side of the insert was thoroughly wiped off using cotton swabs. Emerging lamellipodia in the lower compartment were lysed in a cooled lysis buffer (100 mM Tris pH 7.4, 150 mM NaCl, 5 mM EDTA, 1% NP-40, 0.25% sodium deoxycholate, 1 mM Na_3_VO_4_, 1% SDS and protease inhibitor cocktail). All protein samples were resolved by SDS-PAGE, as indicated in the Western blotting section.

### 2.7. Western Blotting

Total cell protein extracts were prepared using the RIPA buffer. All protein samples were resolved by SDS-PAGE and transferred onto PVDF membranes as previously described by Pelaez et al. [21]. β-Actin or GAPDH were used as Western blot loading controls. Relative protein expression was normalized to β-Actin or GAPDH signal and then to control sample using Gels tool from ImageJ.

### 2.8. Immunoprecipitation 

A549 cell line and A549 S522 clones were cultured to confluence and lysed for 10 min at 4 °C in 1.5 mL cooled IP buffer (1% Triton X-100, 1% NP-40, 50 mM NaCl, 20 mM Tris pH 7.5, 10 mM HEPES, 2 mM EDTA, 1 mM EGTA, 0.1 mM MgCl_2_ and protease inhibitor cocktail). Cell lysates were clarified for 20 min with 1 μg of the appropriate isotype control. Five hundred micrograms of total protein were incubated with 2 μg of anti-EB1, anti-IQGAP1, anti-CRMP2, or anti-pCRMP2 S522 and their corresponding isotype controls for 1 h at 4 °C on an orbital shaker. Afterwards, 20 μL of agarose conjugated immunoreactive protein A (Protein-A/G PLUS-Agarose, Santa Cruz, Dallas, TX, USA) were added and incubated for 16 h at 4 °C in rotation. The protein-antibody-agarose conjugated protein-A complexes were washed three times with a cooled IP buffer and resolved by SDS-PAGE, as indicated in the Western blotting section.

### 2.9. Confocal Microscopy on Fixed Samples

To perform immunofluorescence experiments, 5 × 10^4^ cells were seeded onto 8-well slides (Lab-Teck, Roskilde, Denmark) covered with 50 μg/mL of collagen type I (BD Biosciences, San Jose, CA, USA). Afterward, cells were fixed for 15 min at 37 °C in 4% paraformaldehyde (PFA) (AppliChem, Darmstadt, Germany), and permeabilized for 5 min in 0.5% Triton X-100 (Sigma Aldrich, Steinheim, Germany). Incubation with a specific primary antibody was carried out overnight at 4 °C with the corresponding primary antibodies. Samples were rinsed with PBS and then incubated for 1 h at room temperature with the corresponding Alexa Fluor conjugated secondary antibodies. Notably, CRMP2 and pCRMP2 S522 were fixed with pre-cooled methanol (EGTA 1 mM, MgCl_2_ 1 mM) for 10 min and permeabilized in 0.1% saponin-PBS.

For image acquisition, LSM 510 META inverted confocal microscope (Zeiss, Jena, Germany) equipped with an oil-immersion Plan 63X Apochromatic objective (N.A. 1.40) was used. Images were acquired using the Aim4 software (Zeiss, Jena, Germany). 3D reconstructions and intensity profiles were generated from Z-stacks using the Volocity software (Perkin Elmer, Waltham, MA, USA), and images were analyzed using the free software ImageJ (National Institutes of Health, Bethesda, MD, USA).

### 2.10. Real Time Confocal Microscopy

CRMP2 real-time distribution was analyzed using an Ultraview ERS spinning disk confocal microscope (Perkin Elmer, Waltham, MA, USA) equipped with a water-immersion 63X Plan-Apochromat objective (N.A. 1.40). Briefly, 1 × 10^4^ stably transfected pLEGFP-CRMP2 or pLEGFP cells were placed within a hollow insert on a microwell Petri-dish (MatTek Corporation, Ashland, MA, USA) covered with 50 μg/mL collagen type I. Cells were serum-starved for 16 h. Subsequently, a drop of Matrigel containing 200 ng/mL CCL21 was placed in front of the insert to create the chemotactic gradient. Time-lapse sequences were captured every 2 min for 2 h using the AxioVision software (Zeiss, Jena, Germany). CRMP2 distribution was processed using the Lookup tables tool developed from ImageJ. CRMP2 intensity values per pixel ranging from 0 to 255 (8-bit image). Ten videos were acquired per condition.

### 2.11. Image Analysis

Manders’ overlap coefficient (MOC) to calculate colocalization between fluorescent signals was quantified using the Intensity Correlation Analysis tool developed for ImageJ. A region of interest (ROI) containing the whole cell was selected by the threshold command, and Manders’ overlap coefficient between each channel was analyzed. Non-co-location has an overlap value of zero, while a complete co-location has the maximum overlap value, which is one. The overlap coefficient of at least 50 cells was analyzed for each cell type and condition.

Polarization of CRMP2 and pCRMP2 S522 signal was analyzed using the ImageJ software. To that end, images were binarized using the Moments threshold algorithm. Binary images were processed using a median filter, and the background was removed using the Subtract Background tool with the sliding paraboloid option. Subsequently, the whole-cell area was segmented into two sectors: (i) Sector 1 that comprised half of the cell that contained the leading edge; (ii) Sector 2 that contained the rear edge. Finally, CRMP2 intensity (arbitrary units) was measured in each sector from the obtained masks using the Histogram tool developed for ImageJ. The mean fluorescence intensity (MFI) in Sector 1 and Sector 2 of 50 cells were analyzed for each cell type and condition.

The MTs-membrane contact angle was quantified using the Angle tool developed by ImageJ. Three independent replicas, in which three representative microtubules from 50 cells were randomly analyzed (number of microtubules, n = 450).

The uptake of β1 integrin was assessed by measuring the mean fluorescence intensity (MFI) of β1 integrin (arbitrary units, AU) in 5-micron thick “doughnut” shape ROI in the surrounding area of each cell. The “doughnut ROIs” were selected using the ImageJ Make Band Selection command, specifying the desired thickness at the cell boundary.

### 2.12. Microtubule Regrowth Assay 

To evaluate the microtubule growth rate in the control cell line A549 and the A549 S522 clones, 1 × 10^4^ cells were seeded on 8-well slides (Lab-Tek, Roskilde, Denmark). After cell attachment, cells were serum-starved and incubated with 10 μM nocodazole for 4 h. The wells were rinsed three times with PBS, and microtubule regrowth was stimulated by adding pre-warmed RPMI media supplemented with 10% serum for 1-, 3-, 5-, 10-, and 20 min at 37 °C. Subsequently, cells were either immediately fixed with 4% PFA or lysed in RIPA buffer as described in confocal microscopy imaging or Western blot section, respectively.

Microtubule density and polymerization rate were estimated from confocal images using the ImageJ software. To that end, the tubulin signal was binarized using the Moments threshold algorithm. Binary images were filtered with a median filter, and the background was removed with the Subtract Background tool with the sliding paraboloid option. Finally, the area and mean length of the microtubules from the masks obtained were automatically identified and measured using the Particle Analysis tool. Microtubule density is shown as the percentage of tubulin pixels relative to the whole-cell area. Three independent replicas were analyzed, in which 50 cells per time point (1-, 3-, 5-, 10-, and 20 min) were randomly quantified (number of cells, n = 50). The microtubule polymerization rate is shown as time-normalized mean microtubule length (μm/min) at each time point (1-, 3-, 5-, 10-, and 20 min). Three independent replicas were analyzed, in which 50 cells per time point were randomly quantified (Number of cells, n = 150).

### 2.13. Xenograft Murine Model of Lung Carcinoma 

Nude athymic female mice (20–22 g) aged 5–6 weeks (Harlan Laboratories, Indianapolis, IN, USA) were housed under pathogen-free conditions at the CIMA animal facilities (registration number ES31-2010000132). Animal handling and procedures were in accordance with European (Directive 2010/63/EU) and National (RD-53/2013) legislation, with the approval of the CIMA Animal Experimentation Committees and local Government authorization. Briefly, 33 mice were randomly divided into three groups: (i) n = 13, injected with the A549 wild-type cell line; (ii) n = 11, injected with the A549 S522A clone, and (iii) n = 9, injected with the A549 S522D clone. Mice were subcutaneously injected in the two flanks with 2 × 10^6^ cells in 20 μL of a solution of PBS: Matrigel (1:1). Tumor size was measured every 3–4 days using a caliper. To evaluate tumor growth, tumor volume was calculated using the following formula: V = (A^2^ + B)/2, where V is the volume of the tumor, A is its length, and B is its width. Mice were sacrificed 68 days after the inoculation or when tumors reached a maximal volume of 1 cm^3^ by cervical dislocation, and primary tumors were aseptically removed and fixed in 10% formalin.

### 2.14. Statistical Analysis

Each experiment was performed three times. Data were plotted as mean ± SEM using the GraphPad Prism 5 software. Box and whiskers plots, the box extends from the 25–75th percentile, whiskers show 5–95th percentiles, and the median value is represented as a black square inside each box. All statistical analyses were performed using the SPSS 17.0 software. The normal distribution of data were analyzed using Kolmogorov-Smirnov and Shapiro-Wilk tests. Results were compared by Student’s *t*-test or One-Way ANOVA followed by Bonferroni post-hoc tests. Non-parametric distribution was analyzed using Mann-Whitney U and Kruskal–Wallis tests. Asterisks indicate statistically significant differences between groups (*** *p* < 0.001, ** *p* < 0.01 and * *p* < 0.05).

## 3. Results

### 3.1. CRMP2 Expression Is Needed for Oriented Cell Migration

As a preliminary attempt to assess whether CRMP2 is relevant for lung carcinoma cell migration, we silenced its expression by transiently transfecting A549 lung carcinoma cells with commercially available CRMP2-specific siRNAs (Figure 1A and Appendix A). Please note that, in our experience, it is impossible to stably silence CRMP2 in adenocarcinoma cells since it induces p53 activation and cell death [17]. In CRMP2 silenced A549 cells, we analyzed cell migration through Boyden chambers and invasion through the Boyden chambers covered with Matrigel hydrogels. Furthermore, cell speed (µm/min) and directionality were quantified in confocal microscopy time-lapse videos obtained during wound healing assays. Overall, CRMP2 silenced cells showed reduced migration, and invasive capabilities, and altered migration directionality, compared to their wild-type controls (Figure 1B–F).

Next, we investigated if CRMP2 silencing affects cell adhesion to different extracellular matrix components, e.g., collagen, fibronectin, or vitronectin. Figure 2A demonstrates that CRMP2 silenced cells displayed impaired cell adhesion to collagen I, collagen IV and fibronectin. Integrins are transmembrane adhesion molecules that link the intracellular cytoskeleton to the extracellular matrix, being crucial for the mesenchymal migration of carcinoma cells. Among them, β1 integrin associates with integrins α1 or α2 to form collagen receptors. Thus, we first analyzed by flow cytometry whether CRMP2 silencing modified cell surface expression of integrin β1. As shown in Appendix A, CRMP2 silencing did not affect the amount of integrin β1, as well as its expression on the cell surface. Integrin β1 accumulates into early/recycling endosomes in the upper part of carcinoma migrating cells in a tubulin polymerization dependent manner [22]. Therefore, we decided to study integrin β1 endocytosis in CRMP2-silenced cells by flow cytometry. In this case, after heat-stimulation, we detected significant reduction in integrin uptake in CRMP2 silenced cells (Figure 2B). 

In addition, we analyzed by microscopy the co-distribution of transferrin, a maker of clathrin vesicles, and integrin β1 under the same experimental conditions. We observed co-internalization of both molecules in control conditions, i.e., A549 wild type and A549 cells transfected with a non-target sequence. However, co-internalization of both molecules was significantly lower in CRMP2 silenced cells (Figure 2C and Appendix A). 

These data demonstrate that CRMP2 expression is relevant for the oriented migration of lung carcinoma cells, while its absence compromises the endocytosis of membrane receptors.

### 3.2. CRMP2 Distributes into the Lamella of Migrating Cells

Next, we studied the spatial distribution of CRMP2 during cell migration. To this end, we recorded its cell distribution during wound healing assays using migrating A549 cells transfected with a CRMP2-GFP expressing vector. Representative time-lapse video heat maps (Figure 3A) show how during A549 cell migration, CRMP2 distributes evenly throughout the cytoplasm in non-polarized cells. In contrast, it concentrates in the protruding lamella of the migrating cells, peaking 40 min after cytokine induction. At later times (60–80 min after cytokine addition), CRMP2 gradually disappears from the edge of the cell as the lamella and cell body spread. When a new cellular protrusion appears, around 100 min after cytokine addition, CRMP2 concentrates again at the new leading edge. In contrast, we did not observe GFP enrichment in any cell structure during the migration of the A549 cell line transfected with the GFP empty vector (Appendix A).

To investigate if CRMP2 interacts with tubulin at the leading edge and whether this interaction is governed by phosphorylation, we performed western blot and immunoprecipitation assays on lamellipodia-enriched protein extracts from migrating A549 cells, obtained at different time points (Figure 3B,C and Appendix A). We also studied by confocal microscopy the colocalization between tubulin and CRMP2 during lamella extension and to what extent did it depends on CRMP2 phosphorylation. As shown in Figure 3B,C, we could not detect any CRMP2 phosphorylation 20–40 min after cell polarization at times of lamella extension. In contrast, we detected CRMP2 phosphorylated forms at 60–80 min, being more evident first in Ser522 and later in Thr514, coincident with lamella retraction and CRMP2 disappearance from the cell edge, as shown in Figure 3A. Analysis by confocal microscopy demonstrated that CRMP2 co-distributes with tubulin in the lamella of migrating cells, while the interaction between tubulin and phosphorylated CRMP2 (pCRMP2Ser522) is significantly lower (Figure 3D). This extreme was quantified by measuring Mander’s correlation coefficient (MOC) (Figure 3E). Moreover, we analyzed the distribution of CRMP2 in two different sectors: the advancing lamella (Sector 1) and the rest of the cell (Sector 2) in migrating cells. We observed significant enrichment of CRMP2 at the lamella of migrating cells while phosphorylated CRMP2 at Ser522 was expressed at lower levels and evenly distributed throughout the cell body (Figure 3F). Physical interaction between CRMP2 and tubulin was demonstrated by immunoprecipitation assays using protein extracts obtained from the lamella of migrating cells. In addition, as observed in Figure 3G, CRMP2 co-precipitates with tubulin, while its Ser522-phosphorylated form did not.

From these results, we infer that CRMP2 distributes in the protruding lamella of migrating cells and interacts with tubulin in a phosphorylation-dependent manner.

### 3.3. CRMP2 Phosphorylation Mutants Presented Altered Migration and Invasive Capabilities

Once we had demonstrated that CRMP2 was necessary for the oriented migration of lung adenocarcinoma cells, we wanted to provide genetic evidence on the importance of its phosphorylation. For that purpose, we generated CRMP2 phosphomutant clones by stably transfecting A549 and H1299 NSCLC cells with plasmid vectors expressing CRMP2 Ser522 phosphorylation mimetic (S522D) and phosphorylation-resistant (S522A) mutants. Cell viability and CRMP2 protein expression and phosphorylation were analyzed in CRMP2 mutants. The results (Appendix A) show that the expression of CRMP2 mutants does not compromise the viability of A549 and H1299 NSCLC cells. Next, we characterized the adhesion and migration ability of transfected cells with the same experimental approaches used above. The results shown in Figure 4A, demonstrate that A549 cells expressing the phosphomimetic form of CRMP2 (S522D) present significantly lower (50%) adhesion to ECM proteins than controls. In contrast, cells expressing the phosphodefective (S522A) mutant of CRMP2 increased their cell adhesion to collagen (Figure 4A). Cell migration assays confirmed these results: cells expressing the phosphodefective mutant form of CRMP2 (S522A) experienced higher migration and invasion rates when compared to non-transfected cells, while the CRMP2 S522D phosphomimetic mutant showed reduced cell movement (Figure 4B,C). In addition, cell speed and directionality, calculated by individual cell tracking, was severely compromised in CRMP2 S522D mutants (Figure 4D–F). We also studied centromere orientation by labeling γ-tubulin in the CRMP2 mutants. We observed that the CRMP2 S522D mutant presented lower amounts of oriented centrosomes (Appendix A). Cell speed (µm/min), directionality and MTOC orientation in wound healing assays showed the same course of events: CRMP2 S522A mutants healed the wound inflicted in the cell monolayer at higher speeds than control cells, while CRMP2 S522D phosphomimetic mutants barely advanced from the boundaries of the cell monolayer. The same experiments were performed with a second cell line, the highly metastatic NSCLC cell line H1299, obtaining the same results. Transfected cells expressing the CRMP2 phosphorylation-mimetic mutant displayed reduced cell migration and invasion, while the cell expressing the constitutively desphosphorylated form of CRMP2 (S522A) showed increased migration speed and invasion rates. (Appendix A).

Finally, we asked whether the expression of the CRMP2 mutants altered integrin dynamics. To this aim, we measured integrin β1 internalization by flow cytometry and confocal microscopy in A549 wild-type cells and in A549 cells expressing CRMP2 phosphorylation mutant forms. Integrin β1 internalization in this cell line is mediated by clathrin-coated vesicles and inhibited by chlorpromazine (Appendix A). As shown in Figure 5A–C, A549 cells that express the phosphomimetic forms of CRMP2 (S522D) delayed integrin β1 internalization and recycling to the cell membrane. These results were evident by flow cytometry and confocal microscopy analysis of integrin β1 co-localization with Rab4 and Rab11, two regulatory components of the recycling machinery (Figure 5D–G) [23]. Additional experiments in which we detected by Western blot less phosphorylation of the integrin mediators FAK, Src, and ERK1/2 proteins supported these results (Appendix A). Moreover, CRMP2 phosphomimetic mutants produced less invadopodia-mediated degradation of 3D matrices (Appendix A), consistent with the invasion rates observed in Figure 4C.

### 3.4. CRMP2 Interaction with Membrane Anchoring Proteins Contributes to MT Stability

Microtubules are railroads for cargo transport and vesicle recycling from inner cytoplasmic compartments towards the cell membrane. To reach the cell membrane, MTs need the guidance of +TIP binding proteins, such as EB1. Once there, MTs attach to the cell surface through their interaction with anchor proteins, such as IQGAP1. Since we had observed intense CRMP2 staining at the MT tips, we next asked if the expression of CRMP2 phosphomimetic mutants interfered with MT attachment to the cell membrane through MT binding to EB1 and to the membrane anchor protein IQGAP1.

Analysis by confocal microscopy of the presence of tubulin, CRMP2 and EB1 staining in the different clones showed how CRMP2 preferentially locates behind EB1 in the growing tip of MTs (Figure 6A). Interestingly, clones expressing the phosphorylation-resistant form of CRMP2 (S522A) displayed increased co-staining for EB1 and tubulin protein calculated by the co-staining coefficient (MOC). This in turn suggests increased interaction between these two proteins. Conversely, CRMP2 S522D expressing cells did not show tubulin-EB1 co-staining near the cell membrane and noteworthily, showed higher proportion of MTs bent under the cell membrane than controls (Figure 6B). In addition, we confirmed by immunoprecipitation assays the interaction between CRMP2 and EB-1 in each clone (Figure 6C).

Next, to test whether CRMP2 differential phosphorylation affects tubulin stability, we evaluated CRMP2 protein phosphorylation during MT polymerization. Following what was described for neurons, we observed that CRMP2 is sequentially phosphorylated in Ser522 and Thr514 during tubulin polymerization (Appendix A). After nocodazole treatment, quantification of MTs re-growing rate (μm/min) showed lower MT density and procrastinated tubulin polymerization in cells expressing CRMP2 S522D forms (Figure 6D,E). We proceeded to analyze by Western blot the amount of stable tubulin (detyrosinated-tubulin) in the cytoplasm of CRMP2 clones after performing MT polymerization assays. Results shown in Figure 6F,G showed an apparent reduction in the amount of detyrosinated-tubulin in cells expressing CRMP2 phosphomimetic forms compared to control counterparts.

We further studied the interaction between CRMP2, tubulin and IQGAP1 in the cell membrane by confocal microscopy. As shown in Figure 7A, IQGAP1 distributes in a dotted pattern at the cell membrane of control cells, close to the MTs tip. In these cells, co-localization of CRMP2 and IQGAP at the cellular edge is clearly visible. The analysis of IQGAP1 localization in the CRMP2 phosphorylation-resistant mutants (S522A) rendered more intense staining at the cell membrane. In addition, CRMP2 phosphorylation-mimetic (S522D) mutants showed scarce events of IQGAP1 staining at the cell membrane and lower presence of CRMP2 along the microtubule. When we studied the fluorescence intensities for IQGAP1, CRMP2 and tubulin staining five microns under the cell surface we could not observe any co-signaling for CRMP2 and IQGAP1 in cells expressing phosphomimetic forms of CRMP2. Besides, a high proportion of the MTs bent under the cell membrane (Figure 7B), denoting impaired attachment to the cell surface. We performed immunoprecipitation assays to detect CRMP2 and IQGAP1 interaction using protein extracts from migrating A549 NSCLC cells that confirmed the absence of CRMP2-IQGAP1 interaction in CRMP2 phosphorylation-mimetic (S522D) mutants (Figure 7C).

These results show that CRMP2 interacts with membrane anchoring proteins in phosphorylation dependent fashion and intervenes in microtubule stability.

### 3.5. A549 Lung Carcinoma Cells Expressing CRMP2 Phosphomimetic Mutants form Smaller Tumors Than Wild Type Cells

Finally, we provide in vivo evidence on the participation of CRMP2 in lung carcinoma growth. To that aim, we injected A549 NSCLC cells stably expressing CRMP2 mutants in Ser522 into athymic nude mice and then analyzed tumor growth. As shown in Figure 8, tumors expressing CRMP2 S522A clones grew faster and gave rise to the more extensive primary tumors, while tumors expressing CRMP2 S522D mutants were smaller even than control tumors. These results point at CRMP2 phosphorylation as a druggable target to modulate tumor growth and aggressivity.

## 4. Discussion

Oriented migration is the basis of primary cellular functions such as developmental morphogenesis, tissue repair and tumor metastasis. Cell migration often starts in response to extracellular stimuli such as growth factors or chemokines [24]. As a result, the migrating cell becomes polarized and orients its cytoskeleton and signaling molecules towards the direction of movement. During early cell polarization, microtubules (MTs) are temporarily captured and stabilized near the actin-enriched leading edge of the cell, where they exert the tracking forces necessary to orient the microtubule organization center (MTOC) and the Golgi apparatus. In this process, the coupling between MTs and cortical actin is executed by “plus-end” tracking proteins (+TIPs) such as EB1 and CLASP2 [25]. Several MT-associated proteins (MAPs), such as CLASP, APC, and CRMP2, facilitate microtubule dynamics. Phosphorylation of these proteins by several kinases such as Cdk5 and GSK-3β [26,27] disrupts their binding to MT while inhibition of the same kinases contributes to MT stabilization [28]. 

Deregulation of MT dynamics contributes to cancer progression by endowing cancer cells with invasive and metastatic phenotypes. Invasion and metastasis are hallmarks of cancer and account for 90% of deaths in solid carcinoma patients [29]. Previous findings from our laboratory associated the high phosphorylation of the microtubule interacting protein CRMP2 with poor prognosis in NSCLC patients [17]. In this manuscript, our time-lapse microscopy analysis show that CRMP2 distribution in the lamella of migrating cells occurs in waves. CRMP2 concentrates at the leading edge of the early protruding lamella, where it interacts with tubulin through its non-phosphorylated form. In contrast, at times of lamella retraction, CRMP2 distributes evenly across the cytoplasm and becomes phosphorylated in the Ser522 and Thr514 residues. As a result, CRMP2 interaction with tubulin decreases significantly. This phosphorylation-dependent distribution of CRMP2 at the cellular edge has been described in other cancer cell lines from breast or colon carcinoma patients as well [30]. However, these authors described ROCKII-dependent phosphorylation of the long splicing isoform of CRMP2 (CRMP2L), and contrary to our findings, CRMP2L phosphorylation contributed to cell migration and was associated with a poor prognosis. It would be of interest to determine whether lung cancer cells present this splicing isoform and to what extent it contributes to cell migration. 

Other isoforms of the CRMP2 family of proteins have been associated with lung cancer. CRMP1 was described to act as a tumor suppressor gene in lung cancer. At the same time, the expression of its long splicing isoform LCRPM1 was associated with poor survival and increased metastases to the lymph nodes. This pro-metastatic phenotype was regulated by phosphorylation at Thr628 residue by GSK-3β kinase, which promoted filopodia formation, cell migration, and invasion in cancer cells. The expression of the CRMP1 short form antagonized the pro-tumorigenic role of the long splicing isoform of CRMP1 [31]. Similarly, CRMP4 (DYPSL3) was described as a tumor suppressor in lung cancer. In this case, CRMP4 expression in a small series of stage I and IV carcinoma patients was significantly lower in patients with advanced disease. In addition, the authors demonstrated that CRMP4 silencing promoted metastasis in a mouse model of LLC carcinoma [32]. Finally, the expression of CRMP5 was proposed as a marker of neuroendocrine lung tumors [33]. Significantly, the presence of CRMP5 antibodies have been related to paraneoplastic syndrome associated with small cell lung carcinoma [34]. The specific function of CRMP proteins other than CRMP2 in our model of lung adenocarcinoma has not being studied yet. It may be of interest to dissect whether they operate coordinately during cell migration. In particular, considering that CRMP2 is a cytoplasmic tetrameric protein [35], it may be interesting to study whether it forms heterotetramers with other family members in the context of lung carcinoma.

In our hands, transient silencing of CRMP2 expression in lung adenocarcinoma resulted in impaired oriented cell migration, adhesion, invasion and integrin recycling to the cell membrane. Our results parallel those published by Kaibuchi and collaborators in neurons [36]. These authors demonstrated how CRMP2 silencing impeded the orientation of the axonal growth cone. In addition, transfection of lung adenocarcinoma cells with CRMP2 phosphorylation mutants provided genetic evidence on its importance for cell migration. In our experiments, we demonstrated that constitutive dephosphorylation of Ser522 incremented cell migration. At the same time, cells expressing CRMP2 phosphomimetic forms were unable to migrate directionally.

Cdk5 protein kinase has been proposed as a major player in cancer progression. In various cancers, Cdk5 overexpression correlates with poor prognosis, tumor proliferation, migration, and invasion [37]. In lung cancer, Cdk5 regulates tumor suppressor genes, cytoskeletal remodeling, and immune checkpoints [38]. Therefore, the specific intervention of CRMP2 phosphorylation in lung cancer may be a consequence of its characteristic genetic signature and Cdk5 mutational status. Interestingly, there are specific examples that relate Cdk5-mediated phosphorylation of CRMP2 with oncogenesis [39].

Our data demonstrate how CRMP2 intervention in the migration of lung adenocarcinoma cells is related to its participation in tubulin stabilization at the leading edge. During lamella protrusion, MTs attach to the cell membrane through IQGAP1/APC complexes and form thick MT bundles enriched in +TIP binding proteins that include CLASP170, EB1, and dynein [40]. Our work demonstrates the interaction between CRMP2, IQGAP1, and EB1 in close contact with the cell membrane. Most importantly, in CRMP2 S522D clones, MTs bent under the cell membrane, and IQGAP1 was not present at the cell cortex nor co-stained with tubulin. This outcome follows the same pattern and probably the exact molecular mechanisms described for CLASP2 phosphorylation by GSK-3β [25,41] and suggests that CRMP2 may form part of the same protein complexes. 

Destabilization of MT due to aberrant CRMP2 phosphorylation resulted in defective integrin recycling and in consequence altered cell adhesion and invasion. Stable MTs allow the transport of kinesin and dynein motor proteins carrying vesicles with components of focal adhesion complexes [42]. In this line, CRMP2 regulates the traffic of recycling vesicles by its interaction with dynein in neurons [43]. Our results suggest that aberrant phosphorylation of CRMP2 may impede integrin recycling and consequently tumor attachment to the cell-substrate and cell migration. In fact, in vivo experiments performed in mice demonstrated that A549 cells expressing the CRMP2 S522D phosphomimetic mutant grew significantly slower than their counterparts.

During the last decade, inhibition of CRMP2 has emerged as a new strategy to treat neurodegenerative diseases, chronic pain, and bipolar disorder. Recently, inhibitors affecting CRMP2 ion channel activity, subcellular localization, and phosphorylation have been used in pre-clinical assays [44]. Between them, (S)-lacosamide is a good candidate to be tested in cancer models. This molecule reduces CRMP2 phosphorylation by Cdk5 and GSK-3β and inhibits neurite outgrowth [45]. Moreover, this molecule has been already tested in clinical trials as anti-seizure medication (https://clinicaltrials.gov/ct2/show/NCT02477839 (accessed on 10 October 2021)) further experimentation is thus needed to fully describe the applicability of these drugs in the fight against cancer.

One limitation of the present study is the absence of experimentation in cells with stably silenced CRMP2, in order to avoid any interference caused by the endogenous protein. In our hands, stable silencing of CRMP2 is not possible as it induces p53 activation and apoptosis [17]. Nevertheless, the expression levels of CRMP2 in non-transfected A549 NSCLC cells were significantly lower than in cells transfected to express the CRMP2 mutants. Therefore, while acknowledging the possible presence of small amounts of the CRMP2 endogenous protein, we postulate that the changes observed in migration and MT dynamics of the phosphorylation mutants derived from the expression of the aberrant forms of this protein. This is especially evident in the control experiments performed with non-transfected cells.

In summary, in this article we show for the first time CRMP2 dynamic intervention during lamella extension and MT anchoring to the cellular edge, and postulate its phosphorylation as a candidate target to combat lung cancer progression.

## Figures and Tables

**Figure 1 biomolecules-11-01533-f001:**
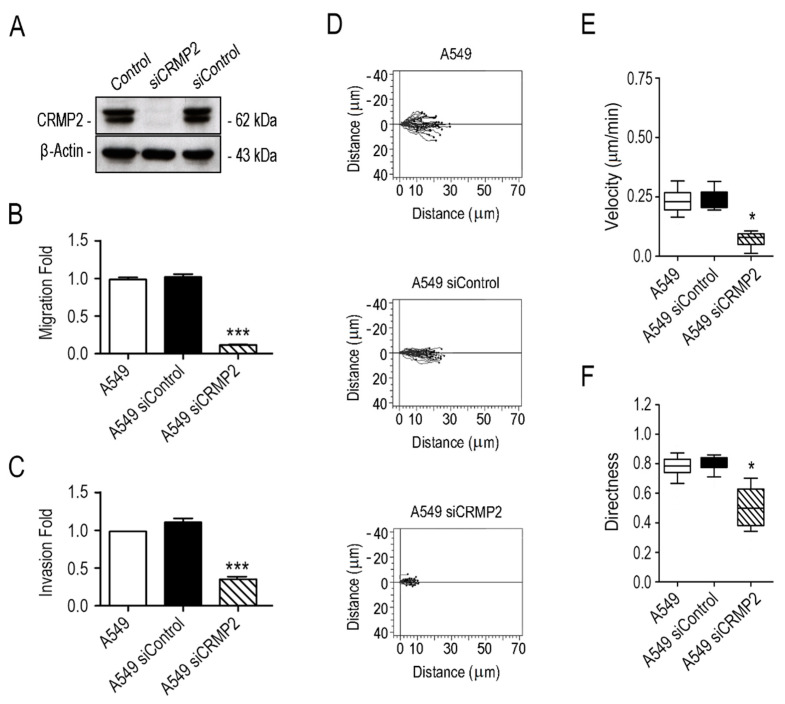
CRMP2 expression is necessary for oriented migration of A549 NSCLC cells. (**A**) Western blot analysis of CRMP2 expression in A549 cells transfected with CRMP2-specific or non-specific siRNA sequences (25 μM). β-Actin was used as a Western blot loading control. (Number of blots, n = 3); (**B**) cell migration towards 20% serum across Boyden chambers. Data are shown as a fold ratio over the migration rate of non-transfected cells. (Number of images, n = 12); (**C**) invasion towards to 20% serum across collagen I covered Boyden chambers (**B**). Data are shown as a fold ratio over A549 wild-type cells. (Number of images, n = 12); (**D**) representative migration tracks of A549 cells transfected with CRMP2-specific or non-target siRNA sequences. Cell migration was captured by confocal video-microscopy every 10 min for 12 h. (Number of videos, n = 3); (**E**,**F**) automated quantification of cell speed (μm/min) and directionality from data obtained in (**D**). (Number of cells, n = 60). *** (*p* < 0.001). * (*p* < 0.05).

**Figure 2 biomolecules-11-01533-f002:**
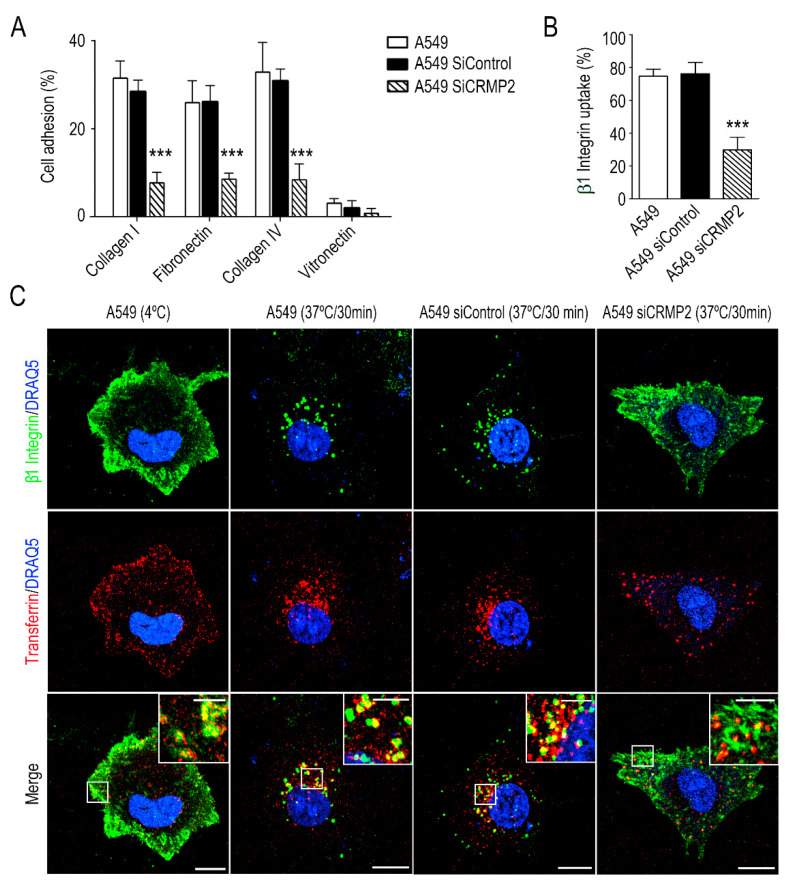
CRMP2 participates in cell adhesion and integrin β1 recycling. (**A**) Cell adhesion of A549 cells transfected with CRMP2-specific or non-target siRNA sequences (25 μM) to collagen I (50 μg/mL), collagen IV (50 μg/mL), fibronectin (10 μg/mL) and vitronectin (1 μg/mL) coated substrates. Data are shown as the percentage of cell adhesion relative to BSA. (n = 12); (**B**) quantification of β1 integrin uptake by flow cytometry in the different cell lines analyzed. Data are shown as the percentage of internalized β1 integrin relative to membrane expression at 4 °C. (n = 3); (**C**) representative maximum intensity projections confocal images of A549 cells stained with β1 integrin (green), transferrin (red), and DRAQ5 (blue). Integrin uptake was stimulated for 30 min at 37 °C in serum-free media. Vesicle internalization was blocked at 4 °C. Scale bar = 15 microns. White insets show magnified areas with strong integrin β1 and transferrin co-localization. Scale bar = 5 microns. *** (*p* < 0.001).

**Figure 3 biomolecules-11-01533-f003:**
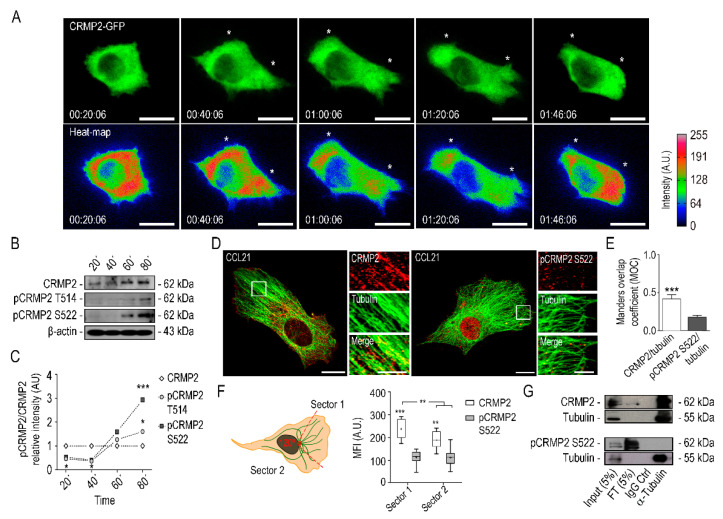
CRMP2 distributes along the lamella of migrating cells. (**A**) Representative confocal images and heat maps of cytoplasmic CRMP2-GFP distribution in A549 cells after CCL21 stimulation (200 ng/mL). Cell migration was captured by confocal video-microscopy with images taken every 2 min for 2 h. Heat maps were rendered using the ImageJ software. White asterisks signal CRMP2 enriched areas at the lamella. Scale bar = 20 microns. (Number of videos, n = 10); (**B**) western blot analysis of CRMP2 and pCRMP2 S522 expression from protein extracts obtained from isolated lamellipodia at the time points indicated. β-Actin was used as a Western blot loading control. (Number of blots, n = 3); (**C**) quantification of CRMP2 and pCRMP2 S522 expression levels from total protein extracts shown in (**B**) (Arbitrary Units, AU). Each sample was normalized to GAPDH expression and, subsequently, to CRMP2 expression; (**D**) representative maximum intensity projections confocal images of actively migrating A549 cells after CCL21 stimulation (200 ng/mL) and stained with specific antibodies to detect tubulin (green), CRMP2 or pCRMP2 S522 (red). Scale bar = 10 microns. White insets show enlarged areas with strong tubulin and CRMP2 or pCRMP2 S522 co-localization at the lamella. Scale bar = 5 microns; (**E**) Manders’ overlap coefficient (MOC) between tubulin and CRMP2 or pCRMP2 S522 positive signal from images shown in (**D**). (Number of images, n = 50); (**F**) quantification of CRMP2 or pCRMP2 S522 distribution in Sector 1 and Sector 2 from images shown in (**D**) following the schema shown on the left. Each cell was segmented into two sectors (120°): Sector 1 comprises the leading edge, and Sector 2 the rear edge. (Number of images, n = 50); (**G**) immunoprecipitation of tubulin and CRMP2 or pCRMP2 S522 from total protein extracts obtained from A549 cells after serum stimulation. (Number of blots, n = 3). *** (*p* < 0.001). ** (*p* < 0.01). * (*p* < 0.05).

**Figure 4 biomolecules-11-01533-f004:**
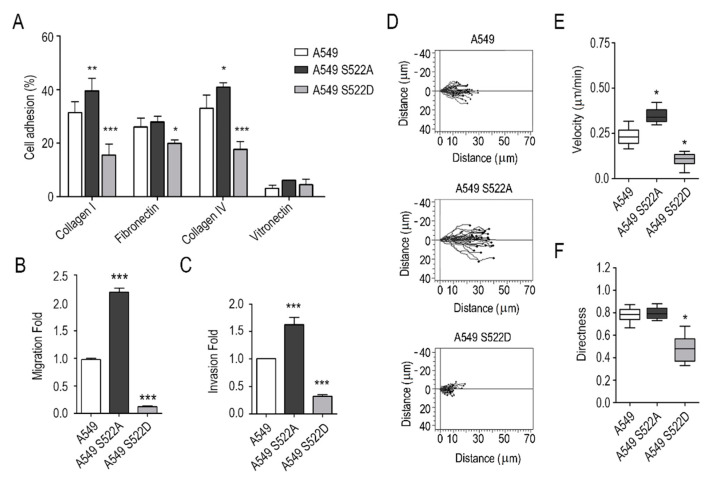
Invasion and adhesion of A549 cells expressing CRMP2 phosphorylation mutant forms. (**A**) Cell adhesion to collagen I (50 μg/mL), collagen IV 50 μg/mL), fibronectin (10 μg/mL) and vitronectin (1 μg/mL) coated surfaces. Results are shown as the percentage of cell adhesion relative to BSA. (Number of wells, n = 12); (**B**) Boyden migration assays towards cell media containing 20% FBS. Data are shown as a fold ratio over wild-type A549 cells. (Number of images, n = 12); (**C**) cell invasion assays across Matrigel hydrogel coated Boyden chambers. Data are shown as a fold-ratio over wild-type A549 cells. (Number of images, n = 12); (**D**) single representative cell tracks of cell movement during wound-healing assays. Cell migration was captured by confocal video-microscopy every 10 min during 12 h. (Number of videos, n = 3); (**E**,**F**) quantification of cell speed (μm/min) and directness from videos shown in (**D**) using the Chemotaxis and Migration software. (Number of cells, n = 60). *** (*p* < 0.001). ** (*p* < 0.01). * (*p* < 0.05).

**Figure 5 biomolecules-11-01533-f005:**
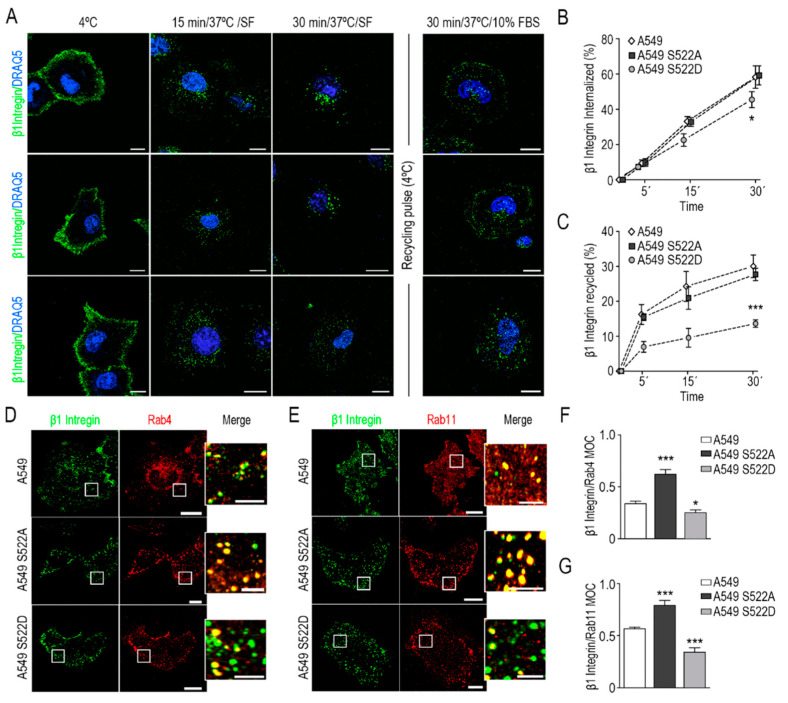
Constitutive phosphorylation of CRMP2 prevents β1 integrin recycling. (**A**) Representative confocal images depicting maximum intensity projections of cells stained with anti-β1 integrin antibody (green) and DRAQ5 (blue). Integrin β1 uptake was stimulated at indicated time points (15 and 30 min) at 37 °C in serum-free media. Integrin β1 recycling cells were exposed to a pulse of cold at 4 °C and immediately stimulated at 37 °C in complete cell culture media for 30 min. Vesicle internalization was blocked by exposure of cell to 4 °C; (**B**) detection of β1 integrin uptake by flow cytometry at the indicated time points (5-, 15-, and 30 min). Data are shown as the percentage of internalized β1 integrin relative to its membrane expression at 4 °C. (n = 3); (**C**) quantification of the rate of β1 integrin recycling by flow cytometry at 5-, 15-, and 30 min after serum stimulation. Data are shown as the percentage of cell surface β1 integrin relative to the percentage of intracellular β1 integrin 30 min after exposure to serum-free media at 37 °C; (**D**,**E**) representative confocal images of maximum intensity projections after staining with anti-β1 integrin (green) and anti-Rab4 or anti Rab11 (red) antibodies. Integrin uptake was stimulated for 5 or 15 min at 37 °C in serum-free media. Scale bar = 10 microns. White insets show enlarged areas with evident co-localization. Scale bar = 5 microns; (**F**,**G**) Manders’ overlap coefficient (MOC) between β1 integrin and Rab4 or Rab11 positive signal from images shown in (**D**,**E**), respectively. (Number of cells, n = 100). *** (*p* < 0.001). * (*p* < 0.05).

**Figure 6 biomolecules-11-01533-f006:**
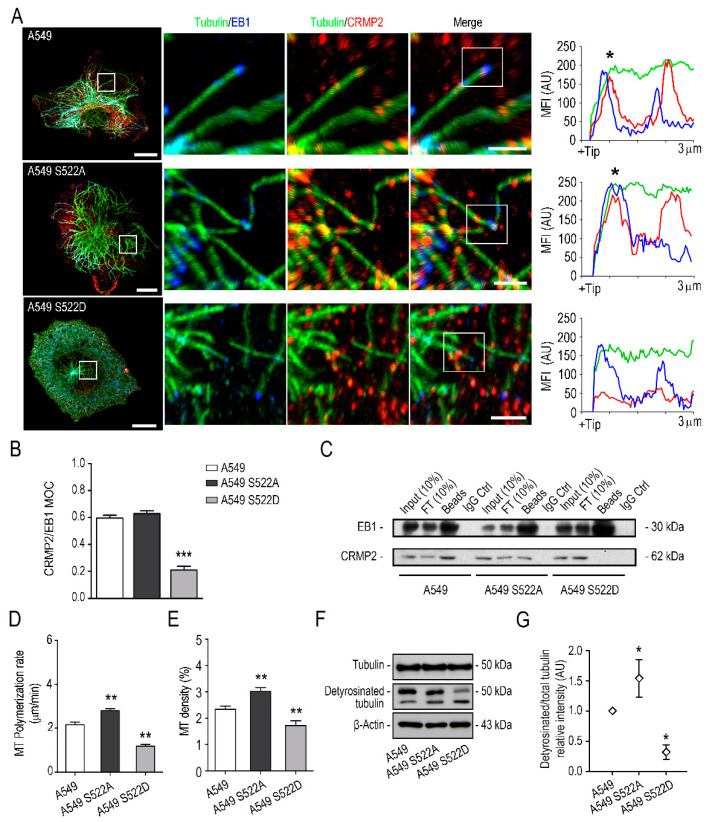
CRMP2 phosphorylation in Ser522 affects binding to EB1 and microtubule stability. (**A**) Representative maximum intensity projections of confocal images of the control cell line A549 and the A549 S522 clones stained with anti-tubulin (green), anti-CRMP2 (red), and anti-EB1 (blue) antibodies 4 h after nocodazole treatment (10 μM) followed by serum stimulation. Scale bar = 10 microns. White inserts show enlarged areas of the +TIP of individual microtubules. Scale bar = 5 microns. Representative fluorescence intensity profiles on the right show strong co-localization between CRMP2 and EB1 (black asterisks) in the first 3 microns of the +TIP of a single microtubule; (**B**) Manders’ overlap coefficient (MOC) between tubulin and EB1 positive signal from images shown in (**A**). (Number of images, n = 50); (**C**) immunoprecipitation of CRMP2 and EB1 proteins from extracts obtained from A549 wild-type and A549 cells expressing the CRMP2 mutants. (Number of blots, n = 3); (**D**) quantification of MT polymerization rate in the different cells after nocodazole treatment (10 μM, 4 h) and subsequent stimulation in 10% serum for 1-, 3-, 5-, 10-, and 20 min. Data are shown as mean MT length per time (μm/min). (Number of cells, n = 150 per time point); (**E**) quantification of MT density in cells treated as in (**D**). Data are shown as the area of tubulin signal relative to the area of the cell. (Number of cells, n = 50 per time point). (**F**) Western blot analysis of tubulin and detyrosinated tubulin protein expression in CRMP2 S522 clones. β-Actin was used as a Western blot loading control. (Number of blots, n = 3); (**G**) quantification of tubulin and detyrosinated tubulin expression levels from total protein extracts shown in (**F**) (arbitrary units, AU). Each sample was normalized to β-Actin expression and, referred, to the expression of tubulin in the A549 control cell line. *** (*p* < 0.001). ** (*p* < 0.01). * (*p* < 0.05).

**Figure 7 biomolecules-11-01533-f007:**
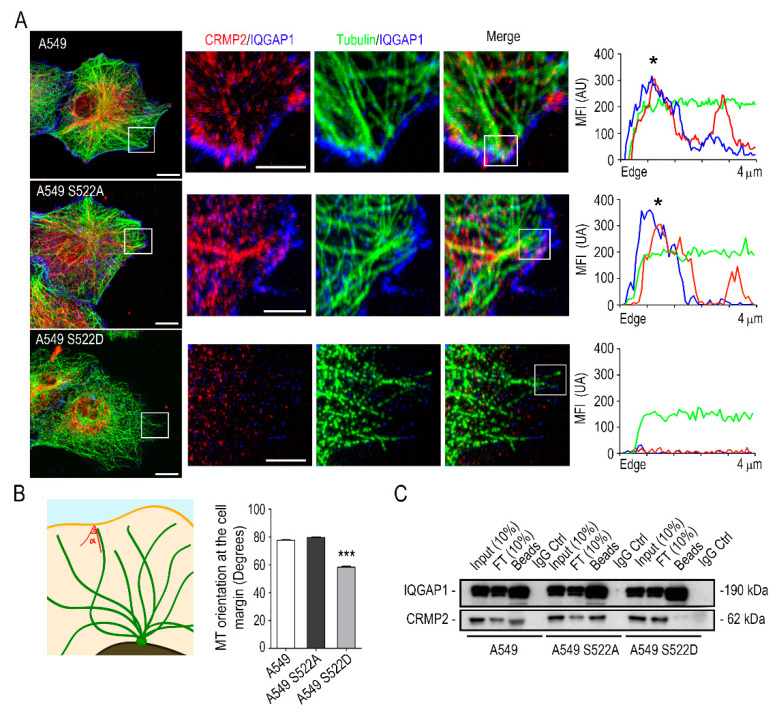
CRMP2 interacts with IQGAP1 at the membrane edge. (**A**) Representative confocal images of maximum intensity projections in the control cell line A549 and in the CRMP2 clones migrating towards CCL21 gradients (200 ng/mL) after being stained with tubulin (green), CRMP2 (red), and IQGAP1 (blue). Scale bar = 10 microns. White insets show enlarged areas of the microtubule docking regions at the cell membrane. Scale bar = 5 microns. Representative fluorescence intensity profiles on the right show the localization of CRMP2 and IQGAP1 (black asterisks) signals in the first 4 microns of the +TIP of a single microtubule; (**B**) quantification of microtubule-membrane contact angle from images shown in (**A**). (Number of microtubules, n = 450). The schematic representation on the left shows the strategy followed to measure the MT-cell membrane contact angle; (**C**) immunoprecipitation of CRMP2 and IQGAP1 proteins from extracts obtained from A549 wild-type and A549 cells expressing the CRMP2 mutants. (Number of blots, n = 3). *** (*p* < 0.001).

**Figure 8 biomolecules-11-01533-f008:**
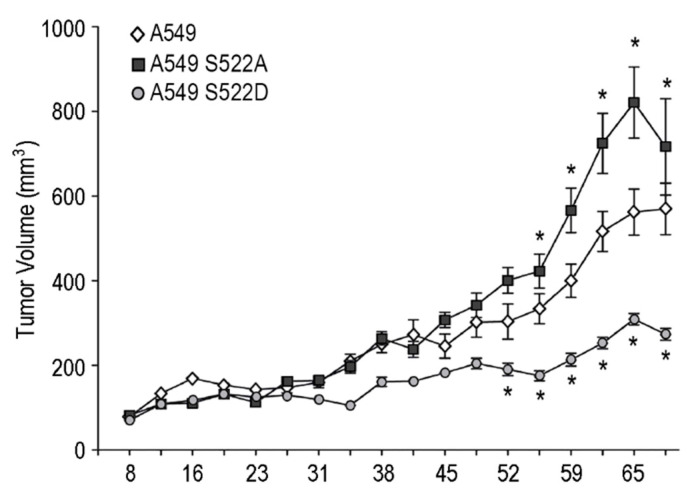
A549 lung carcinoma cells expressing CRMP2 phosphomimetic mutants originate smaller tumors than wild type cells. Tumor growth in nude athymic female mice subcutaneously injected with the control cell line A549 and the A549 S522 clones. Tumor size was measured every 3–4 days. Data are shown as the volume of the primary tumors, calculated as V = (length^2^ + width)/2. * (*p* < 0.05).

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
