# Peer review of "CRMP2 as a Candidate Target to Interfere with Lung Cancer Cell Migration"

_biomolecules, 2021, doi:10.3390/biom11101533_

Round 1
Reviewer 1 Report
The manuscript entitled "CRMP2 as a candidate target to tackle lung cancer cell migration" by Morales et al. provides an exciting dimension to their previously published work. The manuscript is well written, and experiments are adequately planned. My general comments are as follows –
Did the authors observe any variability in the cell proliferation for A549 cells transfected with CRMP-2 Ser522 phospho-site mutants?
What is the role of endogenous wtCRMP2 on migration and invasion when the cells were stably transfected with the mutants CRMP2 (S522A or S522D)? Furthermore, what controls experiments were performed to evaluate the impact of endogenous CRMP2.
Do authors expect similar observations if Thr514 was mutated to Ala or Asp along with Ser522 or just independent of Ser522? Discuss.
Did the authors explore any correlation between Neurofibromin expression and phospho-CRMP2 522 mutants?
Also, the role of other CRMPs, specifically CRMP1 and CRMP4, needs to be discussed in the context of NSCLC.
Abstract – Line 33 – Small molecule inhibitors of CRMP2 phosphorylation are used in clinical trials for neurodegenerative disorders and not in the case of cancer. This sentence can be removed from the abstract as it is misleading and creates confusion regarding CRMP2 phosphorylation and cancer correlation.
The methods section seems to be lengthy, and some sections can be referred back to the earlier publication (ref 17?) or moved to the supplementary section.
Correct the number of cells plated for different experiments, e.g., page 3, line 144 - 1X104 (4 - superscript)
Results section
Figure labelling lacks consistency. Thr514 or T514; Ser522 or S522
Fig 1A, 3B, C, Sup fig 4 F– Include densitometry data to demonstrate quantitative changes for the gels.
Fig3D – Tre514?
Fig 6 and 7 – white insets (or inserts?) appear to be blurred and pixelated. Higher resolution images are required for a clear demonstration.
Fig S5 – Scale bar detail is missing from the legend to figure S5.
Author Response
Comments and Suggestions for Authors
The manuscript entitled "CRMP2 as a candidate target to tackle lung cancer cell migration" by Morales et al. provides an exciting dimension to their previously published work. The manuscript is well written, and experiments are adequately planned. My general comments are as follows –
Thank you very much for your comments regarding this manuscript. They have helped us to improve the clarity and strength of the results presented. Please find enclosed our per-point response to each of the issues raised by the reviewers. In addition, to address these issues we have modified the text -marked in the manuscript- the original figure 3 (modified), 6 (panel 6G) and the supplementary figures: 1 (new figure), 2 (panel C), 3 (panel 3B-C), 4 (panels A-C), 5 (new figure), and 6 (panel 6G).
We hope you find that the issues have been correctly addressed.
- Did the authors observe any variability in the cell proliferation for A549 cells transfected with CRMP-2 Ser522 phospho-site mutants?
We performed proliferation assays using the A549 and H1299 cell lines transfected with the stable CRMP2 constructs. In these experiments, we did not observe any significant variation in their proliferation capacity, as measured by the phenol-red assay. The results of these control experiments have been included as supplementary figures 4C and 5A.
-What is the role of endogenous wtCRMP2 on migration and invasion when the cells were stably transfected with the mutants CRMP2 (S522A or S522D)? Furthermore, what control experiments were performed to evaluate the impact of endogenous CRMP2?
Since we started our research on the role of CRMP2 in lung cancer more than ten years ago, we have tried to produce cells in which CRMP2 could be stably silenced, in order to study the role of the endogenous CRMP2 protein. However, as we published in 2013 (Oliemuller et al, 2013, doi: 10.1002/ijc.27881), this is not possible since CRMP2 silencing induces p53 activation and cell apoptosis several days after siRNA transfection. For this reason, we can only provide experimental data in which CRMP2 has been transiently silenced, for a maximum of 72 hours after siRNA transfection.
Moreover, we have looked for total and phosphorylated levels of CRMP2 protein expression by Western blot in wild-type A549 cells after stably transfection with the CRMP2 mutants. To this end, protein extraction was performed more than ten passages post-transfection. As shown in supplementary figure 4A, non-transfected A549 NSCLC cells express significantly lower levels of CRMP2 than those transfected to express the CRMP2 mutant forms. From this data, we can infer that the changes observed in the migratory behavior and microtubule stability observed with the CRMP2 phosphorylation mutants derive from the expression of aberrant forms of this protein in the presence of a lower amount of the endogenous form of CRMP2. This issue is especially evident in the control experiments that have been performed with non-transfected cells.
-Do authors expect similar observations if Thr514 was mutated to Ala or Asp along with Ser522 or just independent of Ser522? Discuss.
CRMP2 phosphorylation on Thr514 is catalyzed by GSK-3β kinase. For this to occur, CRMP2 must have been previously phosphorylated at Ser522 position by Cdk5, which acts as a priming residue. Therefore, one may speculate that the consequences of mutation in any of the two residues would be similar. However, in our opinion, this is would not the case here since GSK-3β kinase phosphorylates CRMP2 also at Thr509 and Ser518 residues, thus affecting its affinity for tubulin dimers. As it was established in the literature, CRMP2 affinity for tubulin depends on the degree of phosphorylation in the abovementioned residues (Gu et al. 2000, doi: 10.1021/bi992323h). Therefore, to obtain similar effects to Ser522 mutation, it would be necessary to mutate all the GSK-3β residues susceptible of being phosphorylated at the same time as the mutation of a single residue leaves two residues that can be phosphorylated by this enzyme and mask the effect of a single mutation.
In order to provide further biochemical evidence on the role of CRMP2 phosphorylation at residue T514, we treated non-transfected A549 cells with the GSK-3β kinase inhibitor 10 µM LiCl and separately with the Cdk5 inhibitor 10 µM roscovitine for 4 hours at 37°C in serum-free media. With these treatments we expected to inhibit the phosphorylation of CRMP2 at the residue Thr514 and Ser522, respectively. As seen in the figure above treating with these inhibitors significantly reduced the phosphorylation levels of their targeted residues. Furthermore, when we analyzed cell migration of these same cells, we observed significant increments in migration and velocity (µm/min) of the A549 cell line compared to that of non-treated cells, being higher in the case of the cells treated with roscovitine. We provided these results for reviewer’s inspection.
- A) Western blot analysis of CRMP2 and pCRMP2 (T514 and S522) expression in A549 cells after LiCl (10µM) and roscovitine (10µM) treatment for 4 hours. β-Actin was used as a Western blot loading control. (Number of blots, n = 3). B) Quantification of CRMP2 and pCRMP2 (T514 and S522) expression levels from total protein extracts shown in H using the ImageJ software (Arbitrary Units, AU). Each sample was normalized to β-Actin expression and, subsequently, to the expression of CRMP2 in the A549 control cell line. C) Cell migration across Boyden chambers to 20% serum after LiCl (10µM) and roscovitine (10µM) treatment. Data is shown as a fold ratio over A549 non-treated cells. (Number of images, n = 12). D) Quantification of A549 cell speed (μm/min) after LiCl (10µM) and roscovitine (10µM) treatment using the Chemotaxis and Migration software. (Number of cells, n = 60).
-Did the authors explore any correlation between Neurofibromin expression and phospho-CRMP2 522 mutants?
Initially described in neurite outgrowth, Neurofibromin (NF1) regulates CRMP2 phosphorylation via direct and indirect associations with CRMP2. Indeed, Neurofibromin interacts directly with the active (non-phosphorylated) form of CRMP2 through its C-terminal domain but not with the inactive (phosphorylated) form. The interaction between these proteins has been thoroughly studied in the context of pain response, where CRMP2 interacts with Neurofibromin, syntaxin, and voltage-gated ion channels (Moutal et al. 2018, doi: 10.1016/j.neuroscience.2018.04.002).
In this work, we addressed the interaction between CRMP2 and tubulin in NSCLC migration. Although it may be of interest, we have not explored here the correlation between these two proteins in A549 CRMP2 mutants what could very well constitute a future piece of work. In fact, the evidence on the role of Neurofibromin in the onset or biology of lung cancer is relatively recent. The Cancer Genome Atlas Research Network published in 2018 a comprehensible prolife of lung carcinoma in which NF1 was postulated as a driver gene significantly enriched in samples lacking oncogene mutations (doi: 10.1038/s41586-018-0228-6). Already some years before, in 2016 Amanda Redig performed NGS in a panel of 591 lung cancer samples and pointed at the presence of NF1 mutations as a descriptor of unique NSCLC populations (Redig et al. 2016; doi: 10.1158/1078-0432.CCR-15-2377). In addition, NF1 has been studied as a biomarker of resistance in NSCLC (Hongge et al. 2020; doi: 10.3892/etm.2020.8418). Therefore, the study of the relationship between Neurofibromin and CRMP2 in cancer could be indeed of great interest.
-Also, the role of other CRMPs, specifically CRMP1 and CRMP4, needs to be discussed in the context of NSCLC.
Thank you very much for this new insight. This subject has been now properly discussed and included in the manuscript.
In addition to CRMP2, other isoforms of this family of proteins have been associated with lung cancer. CRMP1 was described to act as a tumor suppressor gene in lung cancer. In contrast, the expression of its long splicing isoform LCRPM1 was associated with poor survival and increased metastases to the lymph nodes. Mechanistic studies demonstrated that its pro-metastatic phenotype is regulated by GSK-3β kinase -mediated phosphorylation at Thr628 residue, which promotes filopodia formation, migration, and invasion in cancer cells. This pro-tumorigenic role was antagonized by the short CRMP1 isoform (Wang et al, 2012, doi: 10.1371/journal.pone.0031689). Similarly, CRMP4 (DYPSL3) was described as a tumor suppressor in lung cancer. In this case, CRMP4 expression was compared in a small series of stage I and IV carcinoma patients to find significantly lower CRMP4 expression in patients with advanced disease. Evidence provided by Wang and co-workers in a mouse model of LLC carcinoma showed that CRMP4 silencing promoted metastasis (Yang el al, 2018, doi: 10.1186/s12931-018-0740-0). Finally, the expression of CRMP5 has been proposed as a highly specific marker for neuroendocrine lung tumors (Meyronet et al, 2008, doi: 10.1097/PAS.0b013e31817dc37c). Importantly, CRMP5 auto-antibodies have been associated with paraneoplastic syndrome associated with small cell lung carcinoma (Tolkovsky et al, 2021, doi: 10.1016/j.jneuroim.2021.577635).
CRMP2 is a cytoplasmic protein that forms tetramers. Although its more frequent form is homotetrameric, it can also associate with other CRMP proteins in heterotetramers (Wang et al 2002, doi: 10.1046/j.1471-4159.1997.69062261.x). Whether heterotetramers of CRMP2 with other members of the family occur and are present and active in the context of lung carcinoma has not been studied yet.
Abstract – Line 33 – Small molecule inhibitors of CRMP2 phosphorylation are used in clinical trials for neurodegenerative disorders and not in the case of cancer. This sentence can be removed from the abstract as it is misleading and creates confusion regarding CRMP2 phosphorylation and cancer correlation.
We have modified this sentence for the sake of clarity.
-The methods section seems to be lengthy, and some sections can be referred back to the earlier publication (ref 17?) or moved to the supplementary section.
We have shortened this section and provided references to previously published work.
-Correct the number of cells plated for different experiments, e.g., page 3, line 144 1X104 (4 - superscript)
Done.
Results section
-Figure labelling lacks consistency. Thr514 or T514; Ser522 or S522
Done.
-Fig 1A, 3B, C, Sup fig 4 F– Include densitometry data to demonstrate quantitative changes for the gels.
Supplementary Figure 1 contains densitometry data corresponding to figure 1A.
In figure 3A, the panels D correspond to densitometry from this figure.
A supplementary figure 4F (now supplementary figure 6F), and additional panel with Western blot densitometry has been added.
-Fig3D – Tre514?
Corrected.
-Fig 6 and 7 – white insets (or inserts?) appear to be blurred and pixelated. Higher resolution images are required for a clear demonstration.
We appreciate your comment regarding image quality. When reviewing the manuscript material submitted, we have indeed noticed that the images, as you point out, were somewhat blurry. This may be due to a resolution of resolution during their insertion in the document. Therefore, we have re-inserted them again in order to improve their quality.
Furthermore, we would like to bring the reviewer’s attention to the fact that the maximum lateral theoretical resolution of our Apochromatic Plan 63x oil immersion objective (NA 1.4) is approximately 0.3 microns. This value is near the diffraction limit and can only be improved using super-resolution microscopy systems not available in our environment. In any event, given that the EB1/CRMP2 and IQGAP1/CRMP2 complexes are approximately 1 micron wide at the MT plus-end (see intensity profiles), and our theoretical resolution (0.3 microns), we consider that the quality of the images is appropriate to show their interaction, considering the non-idealities caused by, e.g., optical aberrations not perfectly corrected in the objective lens, the presence of fluorescence scattering, antibody background or sample autofluorescence. In addition, this specific interaction has been demonstrated in immunoprecipitation assays as shown in Figure 6C and 7C.
-Fig S5 – Scale bar detail is missing from the legend to figure S5.
Corrected.
Reviewer 2 Report
The manuscript presented by Morales et al, titled "CRMP2 as a candidate target to tackle lung cancer cell migration" explores the importance of CRMP2 in NSCLC using A549 cells as an in vitro study model to understand the importance of CRMP2 phosphorylation, Particularly S522, on CRMP2/Tubulin interaction and ist effects on cell migration and adhesion using various cell biology tools. The study presented is conducted soundly and tries to address a clinically relevant problem in NSCLC. The submitted data seems robust. However, few major concerns need to be critically addressed before such a claim can be made. In particular, only one cell line was used throughout the study, and similar was the case with the siRNA used.
1. Authors are advised to use more than one NSCLC cell line and siRNA to validate theirs in vitro findings.
2. The authors must present more data on the stability and expression of CRMP2 S522D or A mutant in A549 cells to clarify the observed phenotype was indeed related to the S522 phosphorylation but not the stability of the CRMP2 itself.
3. At least one of the important assays (as presented in figures 5, 6, or 7) must be validated with a cleaner system using siRNA-resistant clones of CRMP2. In the present understanding, it is difficult to judge the effect of native CRMP2 of the A549 system author used throughout the study.
Minor points:
- It is advisable to show the level of S522D or A from the IP experiments presented in figure 6C and Figure 7C. Else it is difficult to interpret how much noise is coming from WT CRMP2.
2) Additional controls are required for many experiments. Figure 2C requires an image and quantification of data from siRNA CRMP2 cells also.
- As I can see, n= 3 for the data presented in figure 6F, thus it's advised to submit the quantification of this data.
- Term Beads must be replaced with α-tubulin in figure 3H.
Author Response
The manuscript presented by Morales et al, titled "CRMP2 as a candidate target to tackle lung cancer cell migration" explores the importance of CRMP2 in NSCLC using A549 cells as an in vitro study model to understand the importance of CRMP2 phosphorylation, Particularly S522, on CRMP2/Tubulin interaction and its effects on cell migration and adhesion using various cell biology tools. The study presented is conducted soundly and tries to address a clinically relevant problem in NSCLC. The submitted data seems robust. However, few major concerns need to be critically addressed before such a claim can be made. In particular, only one cell line was used throughout the study, and similar was the case with the siRNA used.
Thank you very much for your comments regarding this manuscript. They have helped us to improve the clarity and strength of the results presented. Please find enclosed our per-point response to each of the issues raised by the reviewers. In addition, to address these issues we have modified the text -marked in the manuscript- the original figure 3 (modified), 6 (panel 6G) and the supplementary figures: 1 (new figure), 2 (panel C), 3 (panel 3B-C), 4 (panels A-C), 5 (new figure), and 6 (panel 6G).
We hope you find that the issues have been correctly addressed.
- Authors are advised to use more than one NSCLC cell line and siRNA to validate their in vitro findings.
In order to validate our findings, we have also transiently transfected the lung adenocarcinoma cell line H1299 with the same CRMP2 constructs used for A549 cells. We performed viability, adhesion and invasion assays. All results, provided as Supplementary figure 5, demonstrate that the expression of CRMP2 phosphorylation mutants does not significantly affect cell viability as in A549 cells. Furthermore, the expression of CRMP2 phosphomimetic mutants led to impaired cell adhesion and migration, while the expression of the CRMP2 phosphorylation defective CRMP2 mutant significantly incremented the migration speed compared to the control cells. These results confirm that the alteration of CRMP2 phosphorylation affects NSCLC cell migration and invasion, are provided as supplementary figure 5, and referred to in the corresponding section of the text.
Regarding siRNA expression, we used two different methods to silence CRMP2 expression. Firstly, we used shRNA based on the commercially available vectors (pRetrosuper, Screenic, Netherland). This set up causes only partial CRMP2 silencing and was discarded for our experiments. Next, we moved to using siRNA using ON-TARGETplus siRNA system offered by Horizon with the DharmaFect transfection reagent. This system uses a patented modification pattern for specificity that is combined with the SMARTselection algorithm for efficient target gene silencing. With this set up, siRNA off-targets are eliminated by the development of a dual-strand modification pattern to prevent interaction with RISC and favor antisense strand uptake. Besides, antisense strand seed regions are modified to destabilize off-target activity and enhance target specificity. In addition, specific siRNA sequences were used along with by parallel siRNAcontrol oligos to inspect for non-specific targeting. There are several references in the literature that support its use such as the initial work by Anderson and colleagues (https://rnajournal.cshlp.org/content/14/5/853.full), the work published by Rahajeng and coworkers using this same set of siRNA sequences to silence CRMP2 expression (https://doi.org/10.1074/jbc.C110.166066) and our own previous publication (doi: 10.1002/ijc.27881). The efficacy of CRMP2 silencing in our experiments is demonstrated in figure 1A.
- The authors must present more data on the stability and expression of CRMP2 S522D or A mutant in A549 cells to clarify the observed phenotype was indeed related to the S522 phosphorylation but not the stability of the CRMP2 itself.
We checked for the expression of CRMP2 in our constructs by Western blot and compared it to that of endogenous CRMP2. The results are shown as Supplementary figure 4A and B. Endogenous base-line CRMP2 expression in A549 NSCLC cells is very low. Therefore, when cells are transfected with CRMP2 mutant expression vectors, the increments in CRMP2 protein expression are relevant. For reviewer inspection, we provide an image of CRMP2, pCRMP2 S522 and pCRMP2 T514 protein detection by Western blot in cell lines that have gone through more than ten passages after CRMP2 mutant stable transfection. In any case, even under this circumstance, some, albeit significantly lower, endogenous activity of the protein remains that could somehow soften the effects of the expression of the pCRMP2 mutants.
- At least one of the important assays (as presented in figures 5, 6, or 7) must be validated with a cleaner system using siRNA-resistant clones of CRMP2. In the present understanding, it is difficult to judge the effect of native CRMP2 of the A549 system author used throughout the study.
We are in full agreement with this comment. As stated in the previous answer, it is true that in our system native CRMP2 is present, albeit at very low expression levels. To avoid this, as explained to reviewer #1, from the beginning of our experimentation on CRMP2 more than ten years ago, we tried to produce cells in which CRMP2 could be stably silenced, in order to study the role of the endogenous CRMP2 protein. As we published in 2013 (Oliemuller et al, 2013, doi: 10.1002/ijc.27881), this is not possible, as CRMP2 silencing induces p53 activation and cell death. For this reason, we only provided experimental data with cells in which CRMP2 has been transiently silenced.
With more experimental work, we may try to produce siRNA resistant clones in p53 defective NSCLC cells and study whether the effects are the same. However, in our opinion, although possible, the production of stable silencing clones of CRMP2 will significantly delay the publication of this piece of work. In the meantime, pointing in that direction, we have repeated the experiments with transient expression of CRMP2 mutants in the p53 defective cell line H1299 obtaining the same results.
Minor points:
- It is advisable to show the level of S522D or A from the IP experiments presented in figure 6C and Figure 7C. Else it is difficult to interpret how much noise is coming from WT CRMP2.
At this time, we are unable to discriminate between the levels of S522D or S522A derived from the mutants or the endogenous form of CRMP2 because we do not have any protein tag to discriminate between wild-type and transfected forms of CRMP2. Nevertheless, as stated above, the levels of native CRMP2 protein are very low Furthermore, we detected significant differences in the expression of phosphorylated residues and in functional assays between non-transfected cells and those expressing mutants forms of CRMP2. In consequence, we assume that these differences are derived from the expression of CRMP2 mutant forms.
For this reviewer’s inspection, we provide an image of the RNA and protein expression of CRMP2 and its phosphorylated forms in several cell lines
- A) RT-PCR analysis of CRMP2 mRNA in the indicated cell lines. SH-SY5Y cell line was used as positive control of CRMP2 expression. Each sample was normalized to CRMP2 expression in the NHBE primary cell line. B) Western blot analysis of CRMP2 and pCRMP2 (T514 and S522) expression in the indicated cell lines. β-Actin was used as a Western blot loading control. C-E) Quantification of CRMP2 and pCRMP2 (T514 and S522) expression levels from total protein extracts shown in B using the ImageJ software (Arbitrary Units, AU). Each sample was normalized to β-Actin expression and, subsequently, to the expression of CRMP2 or pCRMP2 (T514 and S522) in the NHBE primary cell line.
2) Additional controls are required for many experiments. Figure 2C requires an image and quantification of data from siRNA CRMP2 cells.
Thank you very much for your comment. Quantification of integrin uptake after CRMP2 silencing was performed by flow cytometry since it is more robust to quantify the expression only on the cell surface. The results obtained were shown in figure 2B. Anyhow, we have included the quantification requested by the reviewer. As shown in Supplementary figure 2C, we have measured the mean fluorescence intensity (MFI) of β1 integrin (Arbitrary units, AU) on the cell surface using “Doughnut” shape ROIs of 5 microns thickness in the surrounding area of each cell.
As I can see, n= 3 for the data presented in figure 6F, thus it's advised to submit the quantification of this data
Done, see panel 6G,
Term Beads must be replaced with α-tubulin in figure 3H
Done,
Round 2
Reviewer 2 Report
I now go through the revised work and the authors' responses to my previous concerns and comments.
The authors addressed most of my concerns, major comments and the minor suggestions I made. However, endogenous CRMP2 remained an issue; The authors justified it with difficulties in achieving complete Knockout. In this case, siRNA mediated knockdown and siRNA resistant clones could be ideal options.
If authors want to publish this without doing such an experiment, they must include limitations of this study where they can consist of few lines and the problems with complete Knockout. This will make aware the readers of the potential issue they can have.
If this is not possible as limitations of the study, they can bring this issue to the discussion section.
With these few lines, I am happy to recommend this work for publication.
Author Response
Thank you for your comments. Again they have helped us to improve clarity.
Again, we agree with the reviewer with the inconvenience of lacking CRMP2 knock-out cell lines. We have been unable to produce them in the NSCLC cell lines tested. As the reviewer suggested, we have included a sentence in the Material and Methods section and a paragraph addressing the limitations of the study at the end of the Discussion.
We hope that this answers the reviewer's concerns.